# NOT ALL FEATURES ARE EQUAL: FEATURE LEVELING DEEP NEURAL NETWORKS FOR BETTER INTERPRETATION

## ABSTRACT

Self-explaining models are models that reveal decision making parameters in an interpretable manner so that the model reasoning process can be directly understood by human beings. General Linear Models (GLMs) are self-explaining because the model weights directly show how each feature contributes to the output value. However, deep neural networks (DNNs) are in general not self-explaining due to the non-linearity of the activation functions, complex architectures, obscure feature extraction and transformation process. In this work, we illustrate the fact that existing deep architectures are hard to interpret because each hidden layer carries a mix of low level features and high level features. As a solution, we propose a novel feature leveling architecture that isolates low level features from high level features on a per-layer basis to better utilize the GLM layer in the proposed architecture for interpretation. Experimental results show that our modified models are able to achieve competitive results comparing to main-stream architectures on standard datasets while being more self-explainable. Our implementations and configurations are publicly available for reproductions†.

## 1 INTRODUCTION

Deep Neural Networks (DNNs) are viewed as back-box models because of their obscure decision making process. One reason that makes deep neural networks hard to interpret is that they are able to magically extract abstract concepts through multi-layer non-linear activations and end-to-end training. From a human perspective, it is hard to understand how features are extracted from different hidden layers and what features are used for final decision making.

In response to the challenge of interpretability, two paths are taken to unbox neural networks' decision learning process. One method is to design verifying algorithms that can be applied to existing models to back-trace their decision learning process. Another method is to design models that "explain" the decision making process automatically. The second direction is promising in that the interpretability is built-in architecturally. Thus, the verification feedback can be directly used to improve the model.

One class of the self-explaining models borrows the interpretability of General Linear Models (GLMs) such as linear regression. GLMs are naturally interpretable in that complicated interactions of non-linear activations are not involved. The contribution of each feature to the final decision output can simply be analyzed by examining the corresponding weight parameters. Therefore, we take a step forward to investigate ways to make DNNs as similar to GLMs as possible for interpretability purpose while maintaining competitive performance.

Fortunately, a GLM model naturally exists in the last layer of most discriminative architectures of DNNs (See appendix A.3 for the reason that the last layer is a GLM layer). However, the GLM could only account for the output generated by the last layer and this output is not easy to interpret because it potentially contains mixed levels of features. In the following section, we use empirical results to demonstrate this mixture effect. Based on this observation, one way to naturally improve interpretation is to prevent features extracted by different layers from mixing together. Thus, we

---

† Public Repo URL annonymized for review purpose-See code folder for detailed implementation

directly pass features extracted by each layer to the final GLM layer. This can further improve interpretability by leveraging the weights of the GLM layer to explain the decision making process. Motivated by this observation, we design a feature leveling network structure that can automatically separate low level features from high level features to avoid mixture effect. In other words, if the low level features extracted by the $k^{th}$ hidden layer can be readily used by the GLM layer, we should directly pass these features to the GLM rather than feeding them to the $k + 1^{th}$ hidden layer. We also propose a feature leveling scale to measure the complexity of different sets of features' in an unambiguous manner rather than simply using vague terms such as "low" and "high" to describe these features.

In the following sections, we will first lay out the proposed definition of feature leveling. We then will illustrate how different levels of features reside in the same feature space. Based on the above observations, we propose feature leveling network, an architectural modification on existing models that can isolate low level features from high level features within different layers of the neural network in an unsupervised manner. In the experiment section, we will use empirical results to show that this modification can also be applied to reduce the number of layers in an architecture and thus reduce the complexity of the network. In this paper, we focus primarily on fully connected neural networks(FCNN) with ReLU activation function in the hidden layers. Our main contributions are as follows:

- We take a step forward to quantify feature complexity for DNNs.
- We investigate the mixture effect between features of different complexities in the hidden layers of DNNs.
- We propose a feature leveling architecture that is able to isolate low level features from high level features in each layer to improve interpretation.
- We further show that the proposed architecture is able to prune redundant hidden layers to reduce DNNs' complexity with little compromise on performance.

The remaining content is organized as follows: In section 2, we first introduce our definitions of feature leveling and use a toy example to show the mixture effect of features in hidden layers. In section 3, we give a detailed account of our proposed feature leveling network that could effectively isolate different levels of features. In section 4, we provide a high level introduction to some related works that motivated our architectural design. In Section 5, we test and analyze our proposed architecture on various real world datasets and show that our architecture is able to achieve competitive performance while improving interpretability. In section 6, we show that our model is also able to automatically prune redundant hidden layers, thus reducing the complexity of DNNs.

## 2 FEATURE LEVELING FOR NEURAL NETWORKS

The concepts of low level and high level features are often brought up within the machine learning literature. However, their definitions are vague and not precise enough for applications. Intuitively, low level features are usually "simple" concepts or patterns whereas high level features are "abstract" or "implicit" features.

Within the scope of this paper, we take a step forward to give a formal definition of feature leveling that quantizes feature complexity in an absolute scale. This concept of a features' scale is better than simply having "low" and "high" as descriptions because it reveals an unambiguous ordering between different sets of features. We will use a toy example to demonstrate how features can have different levels and explain why separating different levels of features could improve interpretability.

### 2.1 A TOY EXAMPLE

We create a toy dataset called Independent XOR(IXOR). IXOR consists of a set of uniformly distributed features $\mathcal{X} : \{(x^1, x^2, x^3)|x^1 \in [-2, 2], x^2 \in [-2, 2], x^3 \in [0, 1]\}$ and a set of labels $\mathcal{Y} : \{0, 1\}$. We use top indices for attributes of this toy example. The labels are assigned as:

$$\begin{cases} y = 1 & x^1 \times x^2 > 0 \wedge x^3 > 0.5 \\ y = 0 & otherwise \end{cases}$$

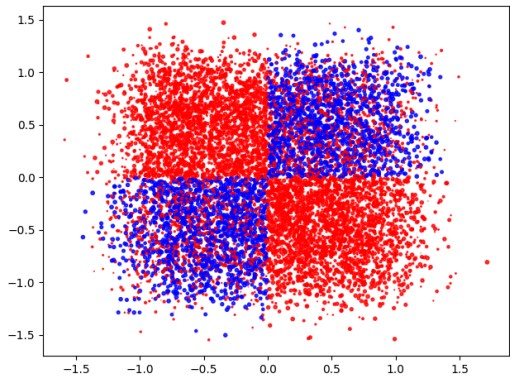

Figure 1: Toy IXOR dataset, with $x^1$, $x^2$ being two coordinates and $x^3$ being the size of the point

In this dataset, $(x^1, x^2, x^3)$ clearly have different levels of feature. $x^3$ can be directly used by the GLM layer as it has a linear decision boundary. $(x^1, x^2)$ is more complex as they form an XOR pattern and cannot be linearly separated, thus requiring further decomposition to be made sufficient for the GLM layer. To make correct decisions, the DNN should use one layer to decompose the XOR into lower level features, and directly transport $x^3$'s value to into the GLM layer.

## 2.2 CHARACTERIZE LOW AND HIGH LEVEL FEATURES WITH FEATURE LEVELING

From IXOR we can see that not all features have the same level of "complexity". Some could be directly fed into the GLM layer, others may need to go through one or more hidden layers to be transformed to features that can directly contribute to decision making.

Thus, instead of using "low" and "high" level to characterize features, we propose to frame the complexity of different features with the definition of feature leveling.

For a dataset $\mathcal{D}$ consisting of $N$ i.i.d samples with features and their corresponding labels $\{(\boldsymbol{a}_1, \boldsymbol{y}_1), ..., (\boldsymbol{a}_N, \boldsymbol{y}_N)\}$. We assume that samples $\boldsymbol{a}_i \in \mathcal{D}$ contains features that requires at most $K$ hidden layers to be transformed to perform optimal inference.

For a DNN trained with K hidden layers and a GLM layer, we define the set of $k^{th}$ level feature as the set of features that requires $k - 1$ hidden layers to extract under the current network setup to be sufficiently utilized by the GLM layer. In the following paragraphs, we denote $\boldsymbol{l}_k \in \mathbb{L}_k$ as the $k^{th}$ level features extracted from one sample and $\mathbb{L}_k$ denotes the set of all $k^{th}$ level feature to be learned in the target distribution. The rest of high level features are denoted by $\mathbb{H}_k$ that should be passed to the $k^{th}$ layer to extract further level features. In this case, $\mathbb{L}_k$ and $\mathbb{H}_k$ should be disjoint, that is $\mathbb{L}_k \bigcap \mathbb{H}_k = \emptyset$. In the case of the toy example, $x^3$ is $\boldsymbol{l}_1$, level one feature, as it is learned by the first hidden layer to directly transport its value to the GLM layer. $(x^1, x^2)$ is $\boldsymbol{h}_1$. The XOR can be decomposed by one hidden layer with sufficient number of parameters to be directly used by the GLM layer to make accurate decisions. Assuming the first hidden layer $f_1$ has sufficient parameters, it should take in $\boldsymbol{h}_1$ and output $\boldsymbol{l}_2$.

## 2.3 HOW THE PROPOSED MODEL SOLVES THE MIXTURE EFFECT AND BOOSTS INTERPRETATION

However, common FCNN does not separate each level of feature explicitly. Figure 2 shows the heatmaps of the weight vectors for both FCNN baseline and proposed feature leveling network trained on the IXOR dataset. We observe from FCNN that $x^3$'s value is able to be preserved by the last column of the weight vector from the first layer but is mixed with all other features in the second layer, before passing into the GLM layer. Our proposed model, on the other hand, is able to cleanly separate $x^3$ and preserve its identity as an input to the GLM layer. In addition, our model is able to identify that the interaction between $(x^1, x^2)$ can be captured by one single layer. Thus, the model

eliminates the second layer and pass $(x^1, x^2)$ features extracted by the first hidden layer directly to the GLM layer.

Looking at the results obtained from the toy example, we can clearly see that the proposed model is able to solve the mixture effect of features and gives out correct levels for features with different complexities in the context of the original problem. Therefore, the model is more interpretable in that it creates a clear path of reasoning and the contirbution of each level of features can be understood from the weight parameters in the GLM.

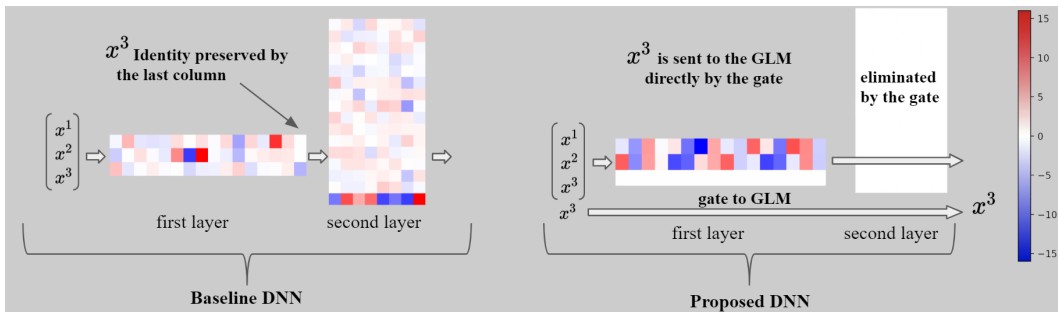

Figure 2: Weight heatmap of Baseline and proposed model with the initial architecture of 3-16-8-2. Arrows denotes information flow. $x^3$ in the proposed model is gated from mixing with other features input to the hidden layer.

## 3 OUR PROPOSED ARCHITECTURE

Inspired by our definition of feature leveling and to resolve the mixture of features problem, we design an architecture that is able to recursively filter the $k^{th}$ level features from the $k^{th}$ layer inputs and allow them to be directly passed to the final GLM layer.

We start with a definition of a FCNN and extend that to our model: we aim to learn a function $\mathcal{F}$ parametrized by a neural network with $K$ hidden layers. The function $\mathcal{F}$ can be written as:

$$\mathcal{F} = d\Big(f_K(f_{K-1}(...f_1(\boldsymbol{a}; \theta_1)); \theta_K)\Big) \tag{1}$$

$f_k$ is the $k^{th}$ hidden layer function with parameters $\theta_k$. $d(\cdot)$ is the GLM model used for either classification, or regression. Thus, the goal is to learn the function $\mathcal{F}$ such that:

$$\mathcal{R}(\theta) = \frac{1}{N}\bigg(\sum_{i=1}^{N} \mathcal{L}(\mathcal{F}(\boldsymbol{a}_i; \theta), \boldsymbol{y}_i)\bigg) \qquad \theta^* = \arg\min_{\theta}(\mathcal{R}(\theta)) \tag{2}$$

In our formulation, each hidden layer can be viewed as separator for the $k^{th}$ level features and extractor for higher level features. Thus, the output of $f_k$ has two parts: $\boldsymbol{l}_k$ is the set of $k^{th}$ level feature extracted from inputs and can be readily transported to the GLM layer for decision making. And $\boldsymbol{h}_k$ is the abstract features that require further transformations by $f_k$. In formal language, we can describe our network with the following equation ("\"denotes set subtraction):

$$\mathcal{F} = d\Big(\boldsymbol{l}_1, \boldsymbol{l}_2, ...\boldsymbol{l}_K, f_K(f_{K-1}(...f_1(\boldsymbol{a}\backslash\boldsymbol{l}_1; \theta_1)) - l_K)\Big) \tag{3}$$

In order for $f_k$ to learn mutually exclusive separation, we propose a gating system for layer $k$, paramatrized by $\phi_k$, that is responsible for determining whether a certain dimension of the input feature should be in $\boldsymbol{l}_k$ or $\boldsymbol{h}_k$. For a layer with input dimension $J$, the gate $\{\boldsymbol{z}_k^1, ...\boldsymbol{z}_k^J\}$ forms the corresponding gate where $\boldsymbol{z}_k^j \in \{0, 1\}$. $\phi_k$ is the parameter that learns the probability for the gate $\boldsymbol{z}_k^j$ to have value 1 for the input feature at $j^{th}$ dimension to be allocated to $\boldsymbol{h}_k$ and $\boldsymbol{l}_k$ otherwise.

In order to maintain mutual exclusiveness between $\boldsymbol{l}_k$ and $\boldsymbol{h}_k$, we aim to learn $\phi_k$ such that the it allows a feature to pass to $\boldsymbol{l}_k$ if and only if the gate is exactly zero. Otherwise, the gate is 1 and the

feature goes to $h_k$. Thus, we can rewrite the neural network $\mathcal{F}$ with the gating mechanism for the $i^{th}$ sample $a_i$ from the dataset:

$$\mathcal{F} = d\Big(B(z_1)\odot a_i, B(z_2)\odot f_1(z_1\odot a_i),...,f_K(z_K\odot f_{K-1}(z_{K-1}\odot f_{K-2}(...f_1(z_1\odot a_i)))))\Big) \quad (4)$$

Here, $\odot$ acts as element-wise multiplication. The function $B$ acts as a binary activation function that returns 1 if and only if the value of $z$ is 1 and 0 otherwise. The function $B$ allows level k feature $l_k = B(z_k) \odot f_{k-1}$ to be filter out if and only if it does not flow into the next layer at all.

Then the optimization objective becomes:

$$\mathcal{R}(\theta, \phi) = \frac{1}{N}\Big(\sum_{i=1}^{N}(\mathcal{L}(\mathcal{F}(a_i, z; \theta, \phi, B), y_i)\Big) + \lambda\sum_{k=1}^{K}||z_k||_0 , \quad z_k = g(\phi_k) \quad (5)$$

With an additional $L_0$ regularization term to encourage less $h_k$ to pass into the next layer but more $l_k$ to flow directly to the GLM layer. $g(\phi)$ act as a transformation function that maps the parameter $\phi$ to the corresponding gate value.

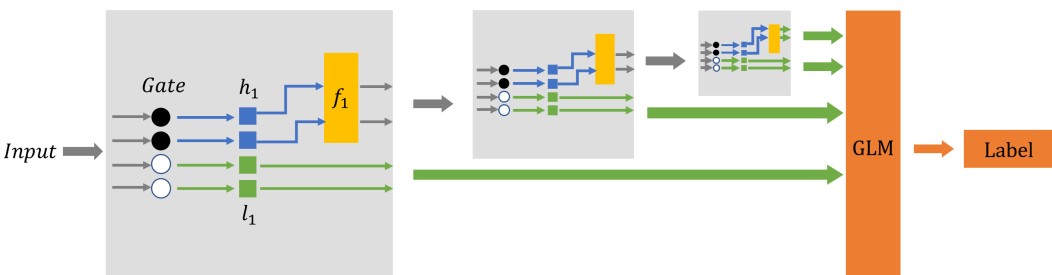

Figure 3: Illustration of the model with three hidden layers. Yellow denotes hidden layer that typically has ReLU activations and green denotes the $k^{th}$ level feature separated out by the gates. Thick arrows denote vector form of input and output. The dimension between the input of the hidden layers and the output can be different.

To achieve this discrete gate construction, we propose to learn the gating parameters under the context of $L_0$ regularization. To be able to update parameter values through backpropogation, we propose to use the approximation technique developed by (Louizos et al., 2017) on differentiable $L_0$ regularization. We direct interested readers to the original work for full establishment of approximating $L_0$ and will summarize the key concept in terms of our gating mechanism below.

Although the gate value $z \in \{0, 1\}$ is discrete and the probability for a certain gate to be 0 or 1 is typically treated as a Bernoulli distribution, the probability space can be relaxed by the following: Consider $s$ to be a continuous random variable with distribution $q(s|\phi)$ paramaterized by $\phi$. The gate could be obtained by transformation function $m(\cdot)$ as:

$$s \sim q(s|\phi), \; z = m(s) = min(1, max(0, s)) \quad (6)$$

Then the underlying probability space is continuous because $s$ is continuous and can achieve exactly 0 gate value. The probability for the gate to be non-zero is calculated by the cumulative distribution function Q:

$$q(z \neq 0|\phi) = 1 - Q(s \leq 0|\phi) \quad (7)$$

The authors furthers use the reparameterization trick to create a sampling free noise $\epsilon \sim p(\epsilon)$ to obtain $s$: $s = n(\epsilon, \phi)$ with a differentiable transformation function $n(\cdot)$, and thus $g(\cdot)$ is equivalent to $m \circ n$ where $\circ$ denotes function composition.

Then the objective function under our feature leveling network is:

$$\mathcal{R}(\theta, \phi) = \frac{1}{N}\Big(\sum_{i=1}^{N}(\mathcal{L}(\mathcal{F}(a_i, z; \theta, \phi, B, g), y_i)\Big) + \frac{\lambda}{K}\sum_{k=1}^{K}\Big(1 - Q(s_k \leq 0|\phi)\Big) \quad (8)$$

$$z_k = g(\phi_k, \epsilon), \quad g(\phi_k, \epsilon) = m \circ n(\phi_k, \epsilon), \quad \epsilon \sim p(\epsilon)$$

## 4 RELATED WORK

**Interpreting existing models:** The ability to explain the reasoning process within a neural network is essential to validate the robustness of the model and to ensure that the network is secure against adversarial attacks (Moosavi-Dezfooli et al., 2016; Brown et al., 2017; Gehr et al., 2018). In recent years, Many works have been done to explain the reasoning process of an existing neural network either through extracting the decision boundary (Bastani et al., 2018; Verma et al., 2018; Wang et al., 2018; Zakrzewski, 2001), or through a variety of visualization methods (Mahendran & Vedaldi, 2015; Zeiler & Fergus, 2014; Li et al., 2015). Most of those methods are designed for validation purpose. However, their results cannot be easily used to improve the original models.

**Self explaining models** are proposed by (Alvarez Melis & Jaakkola, 2018) and it refers to models whose reasoning process is easy to interpret. This class of models does not require a separate validation process. Many works have focused on designing self-explaining architectures that can be trained end-to-end(Zhang et al., 2018; Worrall et al., 2017; Li et al., 2018; Kim & Mnih, 2018; Higgins et al., 2017). However, most self-explaining models sacrifice certain amount of performance for interpretability. Two noticeable models among these models are able to achieve competitive performance on standard tasks while maintaining interpretability. The NIT framework (Tsang et al., 2018) is able to interpret neural decision process by detecting feature interactions in a Generalized Additive Model style. The framework is able to achieve competitive performance but is only able to disentangle up to K groups of interactions and the value K needs to be searched manually during the training process. The SENN framework proposed by (Alvarez Melis & Jaakkola, 2018) focuses on abstract concept prototyping. It aggregates abstract concepts with a linear and interpretable model. Compared to our model, SENN requires an additional step to train an autoencoding network to prototype concepts and is not able to disentangle simple concepts from more abstract ones in a per-layer basis.

**Sparse neural network training** refers to various methods developed to reduce the number of parameters of a neural model. Many investigations have been done in using $L_2$ or $L_1$ (Han et al., 2015; Ng, 2004; Wen et al., 2016; Girosi et al., 1995) regularization to prune neural network while maintaining differentiability for back propagation. Another choice for regularization and creating sparsity is the $L_0$ regularization. However, due to its discrete nature, it does not support parameter learning through backpropagation. A continuous approximation of $L_0$ is proposed in regard to resolve this problem and has shown effectiveness in pruning both FCNN and Convolutional Neural Networks (CNNs) in an end to end manner (Louizos et al., 2017). This regularization technique is further applied not only to neural architecture pruning but to feature selections (Yamada et al., 2018). Our work applies the $L_0$ regularization's feature selection ability in a novel context to select maximum amount of features as direct inputs for the GLM layer.

**Compared to Residual Structures**, our model is able to explain features at different levels and their contributions separately due to the linear nature of GLM. ResNet (He et al., 2016), Highway Networka (Srivastava et al., 2015) cannot isolate each level as their skip features are further entangled by polling, non-linear activation and if the following blocks. Different from ResNet with full connection to all features, we propose to learn which feature to pass to GLM from a probabilistic perspective.

## 5 EXPERIMENTS

We validate our proposed architecture through three commonly used datasets - MNIST, and California Housing. For each task, we use the same initial architecture to compare our proposed model and FCNN baseline. However, due to the gating effect of our model, some of the neurons in the middle layers are effectively pruned. The architecture we report in this section for our proposed model is the pruned version after training with the gates. The second to last layer of our proposed models is labeled with a star to denote concatenation with all previous $l_k$ and the output of the last hidden layer. For example, in the California Housing architecture, both proposed and FCNN baseline start with $13 - 64 - 32 - 1$ as the initial architecture, but due to gating effect on deeper layers, the layer with $32*$ neurons should have in effect $32 + (13 - 10) + (64 - 28) = 71$ neurons accounting for previously gated features. ($13 - 10 = 3$ for $l_1$, $64 - 28 = 36$ for $l_2$).

Table 1: MNIST classification and California Housing price prediction

| MNIST | | | California Housing | | |
|---|---|---|---|---|---|
| Model | Architecture | Accuracy | Model | Architecture | RMSE |
| FCNN | 784-300-100-10 | 0.984 | FCNN | 13-64-32-1 | 0.529 |
| L0-FCNN | 219-214-100-10 | 0.986 | GAM | - | 0.506 |
| SENN (FCNN) | 784-300-100 | 0.963 | NIT | 8-400-300-200-100-1 | 0.430 |
| Proposed | 291-300*-10 | 0.985 | Proposed | 10-28-32* -1 | 0.477 |

The two objectives of our experiments are: 1) To test if our model is able to achieve competitive results, under the same initial architecture, compared to FCNN baseline and other recently proposed self-explaining models. This test is conducted by comparing model metrics such as root mean square error (RMSE) for regression tasks, classification accuracy for multi-class datasets. 2) Because our model can separately account for each layer's contribution, we can apply the gradient with respect to each layer and get the level of features our model recognize for each part of the input.

Experiment implementation details are deferred to appendix A7-10.

## 5.1 DATASETS & PERFORMANCES

**The MNIST hand writing dataset** (LeCun et al., 2010) consists of pictures of hand written digits from 0 to 9 in $28 \times 28$ grey scale format. We use a $784 - 300 - 100 - 10$ architecture for both FCNN baseline and the proposed model. This is the same architecture used in the original implementations of (Louizos et al., 2017). Our model is able to achieve similar result, with less number of layers, as those state-of-the-art architectures using ReLU activated FCNNs . The feature gates completely eliminated message passing to the 100 neuron layer, which implies that our model only need level 1 and level 2 layers for feature extractions to learn the MNIST datasets effectively.

**The California Housing dataset** (Pace & Barry, 1997) is a regression task that contains various metrics, such as longitude and owners' age to predict the price of a house. It contains 8 features and one of the features is nominal. We converted the nominal feature into one-hot encoding and there are 13 features in total. Since California Housing dataset does not contain standard test set, we split the dataset randomly with 4:1 train-test ratio. Our proposed model could beat the FCNN baseline with the same initial architecture. Only 3 out of 13 original features are directly passed to the GLM layer, implying that California Housing's input features are mostly second and third level.

## 5.2 DISENTANGLING THE CONTRIBUTION OF EACH LEVEL OF FEATURES

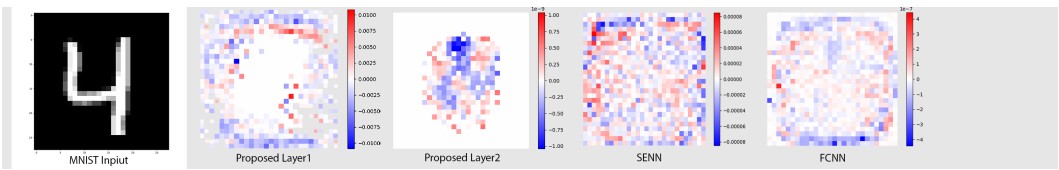

**MNIST:** With digit 4 as an example, compared to FCNN and SENN-FCNN, our model's $l1$ identifies the contour of digit 4 and the corners of 4(larger in gradient) as first level feature for 4. $l2$ shows concentrated negative gradients in the middle of the digit which corresponds to the "hole" in digit 4.

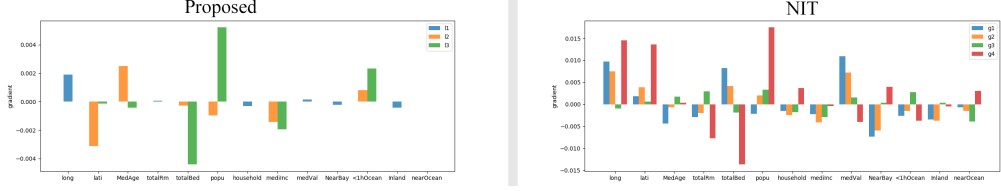

**Cal Housing:** We compare with NIT in [22]. The left figure shows that our model with different colors indicating feature gradients from different layers, NIT's colors indicate different groups.

Compared to [22], our model's $l1$ identifies "longitude"(long) as a feature that linearly relates to housing price since in California, longitude is a major determining factor for housing price comparing to latitude. According to the gradients, $l2$ and $l3$ emphasizes on different parts of the input, justifying that our model could divide the features to different sets. However, for [22], gradients of most groups are similar, indicating that the features are not sufficiently disentangled among groups. In contrast, our model identifies most important features with stronger weight and zero or minimal weight for irrelevant ones.

### 5.3 SCALABILITY

During the training stage, our model requires more computation resources as features from higher layers are passed to the final layer as well as to the final GLM layer. However, during the inference time when the gates are learned, each feature input to the neural layer is only computed once due to the mutual exclusion of our gating setup. The weight parameters related to the "zeroed out" features can also be eliminated. In most cases our model results in lower parameter count. In the Appendix (A2) we show the number of parameters our framework needs for the reported inference models.

### 5.4 EXTEND TO CONVOLUTIONAL NEURAL NETWORKS

Our framework is also applicable to be applied to convolutional architectures. To modify, we simply apply the gate to the input features. Appendix (A11 and Figure 5) shows our model can also clearly isolate features from different levels. To reduce the gated feature size, we apply convolutions with no activation to reduce dimension while maintaining linearity.

## 6 STRENGTH IN PRUNING REDUNDANT HIDDEN LAYERS

Due to our proposed model's ability to encourage linearity, our model is also able to reduce its network complexity automatically by decreasing the number of hidden layers. MNIST classification, as an example, when the dataset feature level is less than the number of hidden layers, our proposed model can learn to prune excess hidden layers automatically as the network learns not to pass information to further hidden layers. As a result, the number of hidden layers are effectively reduced. Therefore, we believe that our framework is helpful for architectural design by helping researchers to probe the ideal number of hidden layers to use as well as understanding the complexity of a given task.

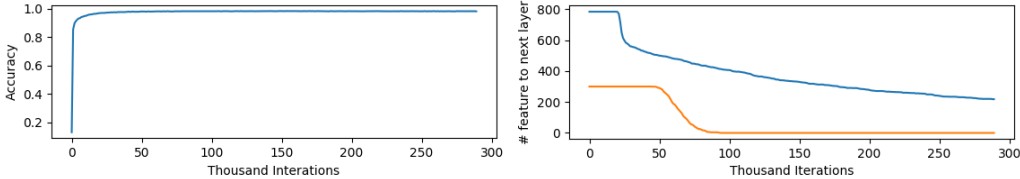

Figure 4: MNIST training performance curve and number of inputs passed to the following hidden layer (blue denotes the number of features passed to the firs hidden layer. Orange curve denotes the second).

## 7 DISCUSSION

In this work we propose a novel architecture that could perform feature leveling automatically to boost interpretability. We use a toy example to demonstrate the fact that not all features are equal in complexity and most DNNs take mixed levels of features as input, decreasing interpretability. We then characterize absolute feature complexity by the number of layers it requires to be extracted to make GLM decision. To boost interpretability by isolating the $k^{th}$ level features. We propose feature leveling network with a gating mechanics and an end-to-end training process that allow the $k^{th}$ level features to be directly passed to the GLM layer. We perfom various experiments to show that our feature leveling network is able to successfully separate out the $k^{th}$ level features without compromising performance.

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

## A   APPENDIX

### A.1   ADDITIONAL EXPERIMENT

Table 2: CIFAR-10 Binary

| Model | Architecture | AUC |
|---|---|---|
| FCNN | 3072-2048-1024-2 | 0.855 |
| GAM (Tsang et al., 2018) | - | 0.829 |
| NIT (Tsang et al., 2018) | 3072-400-400-1 | 0.860 |
| SENN (FCNN) | 3072-2048-1024-2 | 0.856 |
| Proposed | 3072-130- 1024*-2 | 0.866 |

**The CIFAR-10 Dataset** (Krizhevsky et al., 2014) consists of $32 \times 32$ RGB images of 10 different classes. We test our model's ability to extract abstract concepts. For comparison, we follow the experiments in the NIT paper and choose the class cat and deer to perform binary classification.

The resulting architecture shows that for FCNN networks, most of the the two chosen classes are mainly differentiated through their second level features. None of the raw inputs are used for direct classification. This corresponds to the assumption that RGB images of animals are relatively high level features.

## A.2 NUMBER OF PARAMETERS REQUIRED FOR THE REPORTED MODELS

Table 3: MNIST

| Model | Number of parameters |
|---|---|
| FCNN | 266,200 |
| L0-FCNN | 240,962 |
| SENN (FCNN) | 532,400 |
| Proposed | 97,098 |

Table 4: Cal-Housing

| Model | Number of parameters |
|---|---|
| FCNN | 2,912 |
| Proposed | 1,247 |

Table 5: CIFAR-10 Binary

| Model | Number of parameters |
|---|---|
| FCNN | 8,390,656 |
| SENN (FCNN) | 16,781,312 |
| Proposed | 6,430,460 |

Due to unable to reproduce a result reported by NIT paper on CIFAR-10, we used the original architecture that the authors used in the original NIT paper. As a result, we did not include the number of parameters in our table.

## A.3 REVISIT GLM FOR INTERPRETATIONS OF DEEP NEURAL NETWORKS

Consider training a linear model with dataset $\{\mathcal{X}, \mathcal{Y}\}$ where $\mathcal{X}$ is the set of features and $\mathcal{Y}$ is the corresponding set of labels. The goal is to learn a function $f(x)$ from $(x_i, y_i) \in \{\mathcal{X}, \mathcal{Y}\}$ subject to a criteria function $\mathcal{L}_\theta(x_i, y_i)$ with parameter set $\theta$.

In a classical setting of Linear Models, $\theta$ usually refers to a matrix $w$ such that:

$$\hat{y} = f(x) = T(w^\top x + \beta) \qquad (9)$$

Here, $\hat{y}$ refers to the predicted label given a sample instance of a set of feature $x$ and T refers to the set of functions such as Logictic, Softmax and Identity. GLM is easy to interpret because the contribution of each individual dimension of x to the decision output y by its corresponding weight. Therefore, we hope to emulate GML's interpretability in a DNN setting - by creating a method to efficiently back-trace the contribution of different features.

We argue that our proposed architecture is similar to a GLM in that our final layer makes decision based on the weights assigned to each level of input features. Our model is linear in relationship to various levels of features. Given k levels of features, our model makes decision with $y = [w_1^\top l_1, w_2^\top l_2, ..., w_K^\top l_K]$, each weight parameter $w_i$ indicates the influence of that layer. With this construction, we can easily interpret how each levels of feature contribute to decision making. This insight can help us to understand whether the given task is more "low level" or "high level" and thus can also help us to understand the complexity of a given task with precise characterization.

### A.4 The last layer of common neural networks is a GLM layer

The "classical" DNN architecture consists of a set of hidden layers with non-linear activations and a final layer that aggregates the result through sigmoid, softmax, or a linear function. The final layer is in fact similar to the GLM layer since it itself has the same form and optimization objective.

### A.5 Our Novelty Compared to Resnet and Other Similar Architecture

Compared to Residual Networks, our model is able to explain features at different levels and their contributions separately due to the linear nature of GLM. ResNet and DenseNet cannot isolate each level as their skip features are further entangled by polling, non-linear activation and the following hidden layers. Different from ResNet with full connection to all features, we propose to learn which feature to pass to GLM from a probabilistic perspective. Specifically, we introduce the l0 regularization for the purpose of performing effective feature leveling. In contrast, Resnet and Dense Net do not perform such layer wise regularization.

### A.6 Power to extract feature complexity through pruning

To demonstrate that our network achieves effective pruning and can help practitioners to determine the complexity of a given problem, we use Cal Housing as an example and train our models with 2-5 hidden layers. Each intermediate hidden layer has 32-32 structure. To prove that our model can find the optimal structure, we first run the baseline model (without gating) with 2-5 hidden layers separately. We observe that the mse is 0.2364 for 3 layer, 0.2618 for 4 and 0.4807 for 5. Thus, the 3 layer model is sufficient to make accurate prediction. Then we train our model with gate selections and observed that when started with more than 4 hidden layers, our model would completely be reduced to a 3 layer model after training and this is indeed the best structure for the Cal housing task as verified with the complete models. Thus, we argue that our model is able to discover optimal number of hidden layer to make accurate prediction of housing price. This further proves that our model would be helpful for architecture engineers to decide on the optimal number of layers for any given task.

### A.7 Reproducing empirical results: General configuration

All models are implemented in TensorFlow(Abadi et al., 2016) and hyperparameters configurations could be found in our public repository or supplemental code. Model name with citation denotes that the result is obtained from the original paper. SEEN's architecture listed is the prototyping network while we use similar architecture for autoencoder parts. All SENN models are re-implemented with fully connected networks for comparison purposes.

### A.8 Dataset and preprocessing

MNIST is a dataset that contains 60000 training and 10000 testing of handwriting digits from 0 to 9. Experiment results were tested against the allocated testing set.

CIFAR-10 is a dataset consists of 10 classes of images each with 10000 training and 2000 testing. We used the allocated testing set for reporting results.

For MNIST, CIFAR-10, we rescaled the color channel with a divisor of 255., to make pixel values from 0 to 1.

For Cal Housing, we dropped all samples with any empty value entry. Normalize all numerical values with mean and standard deviation.

The IXOR dataset is generated with the script attached in the supplemental material under src/independent_xor.

### A.9 Hyperparameter

The only tunable hyperparameter in our model is the $\lambda$ which we usually consider values from 0.5 to 0.01. All the $\lambda$ values to display result is in the model scripts of the attached folder. Generally, lower

$\lambda$ are better for training more complicated dataset such as CIFAR-10 to prevent too many Gating at early stage.

## A.10 EXACT NUMBER OF ITERATION RUNS

MNIST 280000
CIFAR-10 680000
California Housing 988000

## A.11 RESULT OF OUR MODEL ON CONVOLUTIONAL NEURAL NETWORKS

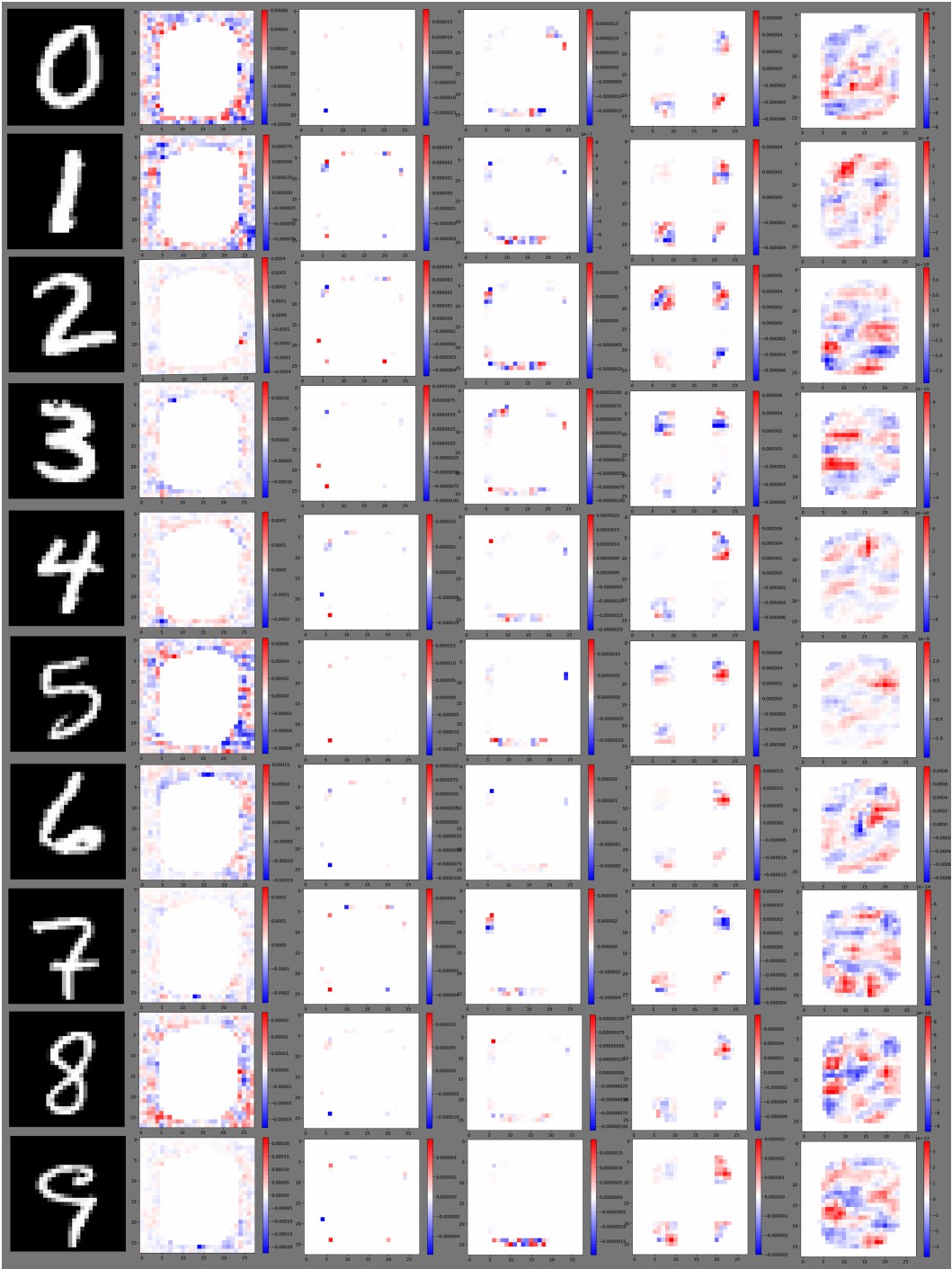

Figure 5: MNIST model trained with 4 layers of convolutional neural network structure. We use the gradients to show different level of features our model extract from each layer. We see that each class is emphasized on features on different levels. 5, 0, 1 and 8 are more sensitive to $l_0$ feature. Some other class are more sensitive to other level of features except from the last level. At the same time, due to feature gating, the highest level feature in the last column is obvious.

