# OpenReview forum: "Not All Features Are Equal: Feature Leveling Deep Neural Networks for Better Interpretation"
_ICLR.cc/2020/Conference — Reject_

### Official Review · AnonReviewer2 · 2019-10-20
**Official Blind Review #2**

**Rating:** 3

**Review:**

This paper proposes a feature leveling technique to improve the self-explaining of deep fully connected neural networks. The authors propose to learn a gated function for each feature dimension for whether to directly send the feature to the final linear layer. The gated function is trained with L0 regularization technique proposed in (Louizos et al., 2017) to encourage more low-level features passed to the final layer. Experimental results on MNIST, California Housing, CIFAR10 show that the proposed method can achieve comparable performance with existing algorithms on sparse neural network training.

Quality:

Overall, the paper is well written with some minor formatting errors. The toy example demonstrates the idea of this paper clearly. However, the novelty of this paper, when compared to NIT, is that the L0 regularization is used to pass the feature to last layer. Considering the self-explaining feature, this work can only explain that some of the input features are suited for final layer, while there is no explanation on the other features since they are used to construct higher level features.

Claririty:

Some parts of this paper are not clear:
1.	Why l_k and h_k has to be disjoint? A feature suited for final classification does not suggest that it can’t be used to construct higher-level feature.
2.	In (4), should not B() be an inverse binary activation function: (1-z)?
3.	Is g(.) the Bernoulli distribution?
4.	In section 5.2, why compare the gradients of a specific input example while one can directly look at z_k, the gated function?
5.	I assume the fully connected layers have bias term. If so, (4) suggests that the gated location will also be added with a learned bias, which is different than what the paper proposes.

Novelty:

The novelty of this paper lies in the sparse training objective becomes passing as many lower-level features to final layer as possible instead of zeros out the intermediate weights. However, the key technique, L0 regularization, has been proposed and used as stated in the related work. While the authors state the application L0 to a novel context to select features is different from prior work, the novelty is rather incremental.

Significance:

This work demonstrates that the L0 regularization technique for sparse neural network training can also be applied to learn a skip-layer connection. However, from both novelty, performance, and self-explaining perspectives, this work does not introduce much to the field.


Pros:

1.	The paper is well written.
2.	The toy example showcases the issue that this work tries to tackle.
3.	The experimental results show the comparable performance to existing works.

Cons:
1.	The novelty is not sufficient considering the prior works on sparse neural network training.
2.	There are some clarification issues as mentioned before.
3.	The performance is only comparable to existing works.
4.	The self-explaining contribution is not clear since only a few input features can be explained if they are passed to the final layers.
5.	There is no experiment on how \lambda would affect the resulting network architecture.


Minor corrections:
1.	First paragraph on sec. 5: three datasets: two datasets (or mention it’s in appendix).
2.	5.2 compare to NIT: the citations are in wrong format. Also the reference for NIT is corrupted.


**Experience Assessment:**

I have published in this field for several years.

**Review Assessment: Checking Correctness Of Derivations And Theory:**

I assessed the sensibility of the derivations and theory.

**Review Assessment: Checking Correctness Of Experiments:**

I assessed the sensibility of the experiments.

**Review Assessment: Thoroughness In Paper Reading:**

I read the paper at least twice and used my best judgement in assessing the paper.

---

### Official Review · AnonReviewer1 · 2019-10-21
**Official Blind Review #1**

**Rating:** 3

**Review:**

This paper proposed a neural network architecture to separate low-level features and high-level features. At the k-th hidden layer, 1) the k-th level features, defined as the set of features that requires k-1 hidden layers to extract, are directly passed to the final GLM layers; 2) the remaining features are further processed in the subsequent layers. This separation is achieved by applying the gating mechanism. The model can be interpreted by the weights associated with each of those k-th level features in the final GLM layer. Experimental results on the MNIST classification dataset and the California Housing regression dataset demonstrate the proposed approach can 1) achieve competitive performance (in terms of classification and regression) compared to the FCNN baseline; and 2) has better interpretability performance.

This paper is clearly written. The description of the model architecture is easy to follow. The introduction of the related works and background material are well organized.

My major concern with this work is how to interpret those k-th level features. Since the weights of those k-th level features in the final GLM model are used to indicate the importance of those features, interpreting the meaning of those k-th level features seem necessary. In practice, how to interpret the meaning of those higher-level features? For example, for the California Housing dataset, how to interpret the meaning of those features learned at level 2?

The interpretation of the MNIST classification examples seems difficult to understand. Compared to inspecting the raw pixels, would it be easier to interpret through learning a few prototypes, similar to the approach described in Alvarez-Melis and Jaakkola NIPS 2018?

**Experience Assessment:**

I do not know much about this area.

**Review Assessment: Checking Correctness Of Derivations And Theory:**

I carefully checked the derivations and theory.

**Review Assessment: Checking Correctness Of Experiments:**

I carefully checked the experiments.

**Review Assessment: Thoroughness In Paper Reading:**

I read the paper thoroughly.

---

### Official Review · AnonReviewer6 · 2019-11-08
**Official Blind Review #6**

**Rating:** 3

**Review:**

Summary:

This paper proposes a novel architecture that is able to separate different types of features learned at each layer of a neural network through a gating structure -- features that are sufficiently passed through the network are immediately sent to the final output layer. In addition, they provide reasonable definitions of levels of features, in contrast to the standard "low" to "high" descriptions. Lastly, in order to make the model more interpretable, they utilize an L0 loss on the gates of each layer to prioritize lower level features being used in the final layer.

Significance:

Although gating is not novel, their use to send kth-level features to the final GLM layer is. Other than that, not much is contributed, as their differentiability trick, as mentioned, has already been done. The motivation to separate different types of features is interesting and definitely an issue that should be studied more.

Quality:

The paper is easy to follow and nicely written, but with a few minor typo issues:

1. Page 1, refers to appendix A.3 but should be for A.4
2. Page 2, "section" is inconsistently capitalized
3. Page 6, mentions three commonly used datasets but only mentions MNIST and California Housing.
4. Page 8, mentions Appendix A.11 for CNN but this section is empty.

In regards to content quality, a few things stand out that could be improved:

1. A major issue is that the interpretability of features with k > 1 are still not explained -- all we know is that they don't need to be sent further through the network. (i.e. solves the separation issue but leaves gaps in interpretability)
2. Since the gates themselves can be studied, rather than finding gradients, wouldn't a simpler way to explain the network be to look at which features are passed to the GLM layer? This would especially be helpful in the first layer when looking at the original input features.
3. Currently it is not clear if the architecture learns when features (l_k) are directly "useful" for classification, or if they are just not compatible with the features passed on to the next layer (h_k).
4. In terms of interpretability, only a few other methods are tested, and gradients are the only way they compare. An exploration of other attribution methods could have further supplemented their claims.
5. Claims are made about how many layers a certain dataset needs for sufficient classification through heuristic experiments; however they are not thorough enough in terms of ablation to fully make this claim. Width of layers are chosen but not analyzed; how is gating affected by the width of the network? For example, in MNIST, would only 3 layers be needed if the width is increased or decreased? This isn't immediately clear.
6. Extensiveness of experiments -- I do like the toy dataset as an example, but to show effectiveness of this framework, a larger breadth of datasets could have been used. As an example, in the SENN paper, they utilize breast cancer and COMPAS but these were not tested on this architecture.  In addition, the results from convolutional layers would be much more preferred, since the best performing architectures on large vision datasets such as ImageNet primarily use convolutions.

**Experience Assessment:**

I have read many papers in this area.

**Review Assessment: Checking Correctness Of Derivations And Theory:**

I carefully checked the derivations and theory.

**Review Assessment: Checking Correctness Of Experiments:**

I carefully checked the experiments.

**Review Assessment: Thoroughness In Paper Reading:**

I read the paper at least twice and used my best judgement in assessing the paper.

---

### Official Review · AnonReviewer7 · 2019-11-12
**Official Blind Review #7**

**Rating:** 1

**Review:**

This paper proposes a categorization of inner layer weights to be linearly or non-linearly correlated with the output. The motivation on why this is important is somewhat weak in the paper. But I could see cases where this is important, if there is supporting evidence that it helps with interpretability. However, I did not see that in this draft unfortunately.

“GLMs … interactions of non-linear activations are not involved” – the link function in all GLMs except linear regression is non-linear. So I am not sure what this statement means.

The definitions of \mathbb{L_k} and \mathbb{H_k} are not clear when they are introduced. The example given talks about l_1 and h_1, not about \mathbb{L_k} and \mathbb{H_k}.

I didn’t get the point of the function B(.). Why not directly use z \in {0,1} ?  Similarly, I did not get the point of introducing g(.), why not just push the entire mapping into \phi ? The notation is made unnecessarily complicated.

The sentences after eq 7 are unclear to me. What does \epsilon \sim p(\epsilon) even mean ? Is \epsilon the parameter or the random variable here ? What is \epsilon used for ? What is m(.) ?

The toy example is simple. But it does not address why such a classification into h_k and l_k is helpful and how it can be used. Can the authors motivate the usecases where such an interpretation (using the toy example) is useful ?

The experiments are not convincing. The MNIST experiment regarding identification of contours is very vague. There are tons of other methods that give better interpretation. The identification of “long” feature to be linearly correlated with the output can be done in a much faster and easier way, by simply checking individual feature correlations. The strength of such a method that the authors are proposing would be to extract useful information for cases where the input features are /not/ linearly correlated, while there are some inner layer features which /are/ linearly correlated. This could help in understanding the landscape of the classification better. But I did not see that happening here. Happy to be proved wrong by the authors or other reviewers if I am missing something here. Similarly, there is lot of work on sparsification/pruning of NNs, in light of which I am not sure what Section 6 adds.

Appendix A.4 is redundant, softmax/sigmoid being a glm is well-known. Also, in the main text (e.g. last para page 1), the reference is to Appendix A.3, while it should be to A.4.


**Experience Assessment:**

I have published one or two papers in this area.

**Review Assessment: Checking Correctness Of Derivations And Theory:**

I carefully checked the derivations and theory.

**Review Assessment: Checking Correctness Of Experiments:**

I carefully checked the experiments.

**Review Assessment: Thoroughness In Paper Reading:**

I read the paper thoroughly.

---

### Decision · Program_Chairs · 2019-12-19

**Decision:**

Reject

**Comment:**

This paper proposes to learn self-explaining neural networks using a feature leveling idea.  Unfortunately, the reviewers have raised several concerns on the paper, including insufficiency of novelty, weakness on experiments, etc. The authors did not provide rebuttal. We hope the authors can improve the paper in future submission based on the comments.